



# Spatial and temporal variability of methane emissions and environmental conditions in a hyper-eutrophic fishpond

Petr Znachor[1,2], Jiří Nedoma[1], Vojtech Kolar[1,3], Anna Matoušů[1]

[1]Biology Centre of Czech Academy of Sciences, v.v.i., Institute of Hydrobiology, Na Sádkách 7, České Budějovice, 37005, Czech Republic

[2]Faculty of Science, University of South Bohemia, Branišovská 1760, České Budějovice, 37005, Czech Republic

[3]Biology Centre of Czech Academy of Sciences, Institute of Entomology, Branišovská 31, České Budějovice, 370 05, Czech Republic

*Correspondence to:* Anna Matoušů (anna.matousu@gmail.com)

**Abstract.** Estimations of methane ($CH_4$) emissions are often based on point measurements using either flux chambers or a transfer coefficient method which may lead to strong underestimation of the total $CH_4$ fluxes. In order to demonstrate more precise measurements of the $CH_4$ fluxes from an aquaculture pond, using higher resolution sampling approach we examined the spatiotemporal variability of $CH_4$ concentration in the water, related fluxes (diffusive and ebullitive) and relevant environmental conditions (temperature, oxygen, chlorophyll-a) during three diurnal campaigns in a hyper-eutrophic fishpond. Our data show remarkable variance spanning several orders of magnitude while diffusive fluxes accounted for only a minor fraction of total $CH_4$ fluxes (4.1–18.5 %). Linear mixed-effects models identified water depth as the only significant predictor of $CH_4$ fluxes. Our findings necessitate complex sampling strategies involving temporal and spatial variability for reliable estimates of the role of fishponds in a global methane budget.

Keywords: aquaculture, emissions, fishpond, freshwater, heterogeneity, methane



## 1 Introduction

Freshwater aquaculture ponds (fishponds) represent man-made counterparts to natural shallow lakes (Scheffer, 2004) which are mainly used for fish production (mostly of common carp, *Cyprinus carpio* L.) and water retention in the landscape. Fishponds serve also as secondary biotope for various organisms (Kolar et al., 2021), supporting noteworthy animal and plant diversity (Pokorný and Hauser, 2002). However, most fishponds suffer from high fish stock densities, excessive carbon and nutrient loading from supplemental fish feeding, sewage pollution, and fertiliser runoffs from agricultural catchments or nutrient mobilisation from the anoxic sediment layers (Pechar, 2000). As a result, the trophic structure of plankton communities has shifted towards a reduction of large zooplankton and massive development of phytoplankton, especially cyanobacterial blooms (Potužák et al., 2007), limiting light penetration in the water column. Rapid changes in the intensity of biological processes such as photosynthesis and respiration often result in pronounced daily or seasonal fluctuations in dissolved oxygen (Baxa et al., 2021), signalling decreasing ecosystem stability. The extent of anoxia, accumulation of organic biomass, and rapid heating of the shallow water during summer result in enhanced production of greenhouse gases (Grasset et al., 2018, Zhang et al., 2021; Bartosiewicz et al., 2021).

Most concerning are $CH_4$ emissions as freshwater aquaculture systems release more than 6 Tg $CH_4$ $yr^{-1}$ (Yuan et al., 2019). Methane can be emitted via several pathways: simple molecular diffusion, ebullition (in the form of bubbles released from oversaturated sediments), plant-mediated flux (Bastviken et al., 2004), but also through so far neglected pathways including aeration, emissions from dry/drying sediments, or dredged organic material (Kosten et al., 2020). Among all, ebullition is considered the dominant pathway (van Bergen et al., 2019; Kosten et al., 2020), which can contribute 50-96 % (Casper et al., 2000; Xiao et al., 2017; van Bergen et al., 2019; Yang et al., 2020; Zhao et al., 2021) to the total $CH_4$ flux. Along with the second important pathway – molecular diffusion, both exhibit high spatiotemporal variability due to various physical and biological factors acting on very short time scales, for instance, temperature (van Bergen et al., 2019), eutrophication (Zhang et al., 2021), water depth (DelSontro et al., 2016), $CH_4$ production rates (Zhou et al., 2019), $CH_4$ oxidation rates (Sanseverino et al., 2013), dissolved oxygen concentration (Xiao et al., 2017), management regime (Yang et al., 2019), or the quality of organic matter in the sediment (Schmiedeskamp et al., 2021). Recently, the direct involvement of phytoplankton in $CH_4$ production and emissions has been emphasised (Yan et al., 2019; Bižić et al., 2020; Bartosiewicz et al., 2021).



Although fishponds are recognised as powerful model systems for studies in ecology and evolutionary or
conservation biology (De Meester et al., 2005; Céréghino et al., 2008), the extent of environmental heterogeneity
in fishponds and shallow Inland small waterbodies remains poorly understood (Ortiz and Wilkinson, 2021), largely
because the driving factors are either system-specific or highly variable on short time scales (Laas et al., 2012).
Most of current information on lentic ecosystem structure and function comes from single-site sampling, in which
measurements are taken over time at the deepest point in the lake, which does not sufficiently account for within-
lake spatial variation (Stanley et al., 2019). The motivation for our study was the growing concern about the role
of fishponds as important sources of $CH_4$ fluxes to the atmosphere (Wik et al., 2016). Unfortunately, the majority
of global $CH_4$ flux estimates rely on upscaling methods (DelSontro et al., 2018a) based on a limited number of
measurements that do not account for diurnal and seasonal variability or ecosystem spatial heterogeneity. Yang et
al. (2019) indicates that a larger number of spatial replicates over a number of months is mandatory to improve
the accuracy of whole-pond $CH_4$ flux estimates. The published research from other aquaculture studies have been
performed mainly in tropical and subtropical zones in fish or crab aquacultures (e.g., Hu et al., 2016; Ma et al.,
2018; Yang et al., 2019, 2020; Yuan et al., 2019, 2021). To better understand the spatial dynamics of $CH_4$ fluxes
and environmental heterogeneity in temperate freshwater shallow lake, we conducted a spatial sampling of the
hyper-eutrophic Dehtář fishpond (Czech Republic, Europe). Since the seasonal $CH_4$ production is strongly affected
by temperature, we focused on warm summer months where the total $CH_4$ fluxes were expected to be the highest
(Jansen et al., 2020). The objectives of our study were (i) to determine the spatial heterogeneity of $CH_4$ diffusive
and total fluxes and fundamental limnological variables (oxygen, temperature, chlorophyll-a) and they change
daily and monthly in the hyper-eutrophic pond, and (ii) to identify the factors that influence $CH_4$ fluxes to improve
our understanding of the importance of spatiotemporal variability for global estimates of $CH_4$ efflux to the
atmosphere.
**2 Material and Methods**
**2.1 Study site description**
The Dehtář fishpond (49° N, 14° E) is a shallow man-made lake (average and maximum depth: 2.4 and 6 m)
constructed in 1479 and used for polycultural, semi-intensive production of common carp (Potužák et al., 2016).
It lies in a flat agricultural landscape at 406.4 m above sea level in the upper Vltava River basin in South Bohemia
(Czech Republic) which is characteristic with its network of fishponds (Fig. 1b). Due to the orography of the





landscape, the Dehtář fishpond, surrounded by narrow belts of littoral vegetation and adjacent to grassland and
arable land, is exposed to wind, mainly from the northwest (for aerial photograph, see Suppl. Fig 1). The catchment
area is 91.4 km$^2$. The main inflow is the Dehtářský stream in the south, while several smaller tributaries flow in
from the west (Fig. 1c). The fishpond has a dam 234 m long and 10 m high, with two outlets and a safety spillway.
Covering 2.28 km$^2$, the Dehtář fishpond is among the ten largest fishponds in the Czech Republic, holding a
volume of $4.71 \times 10^3$ m$^3$ and having a water residence time of 146-445 days (Potužák et al., 2016).

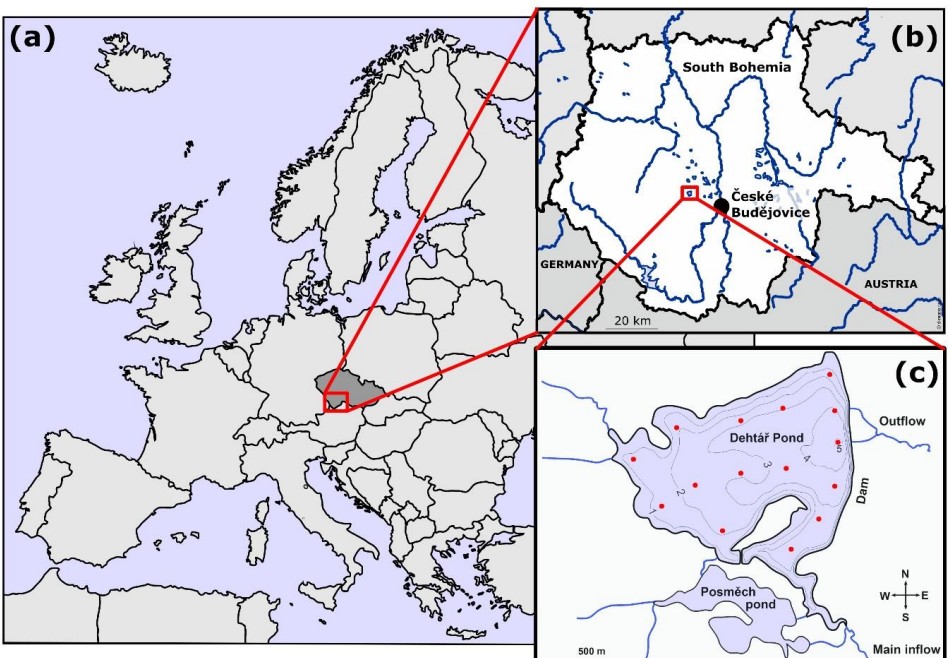


**Figure 1.** Location (a, b; copyright www.d-maps.com; https://d-maps.com/carte.php?num_car=2232&lang=en and https://d-
maps.com/carte.php?num_car=265046&lang=en; modified) and bathymetric map (c; credit Jiří Jarošík) of the sampled Dehtář
fishpond: Blue lines indicate hydrological connections; red dots are the sampling points. Numbers indicate isobath depth.

**2.2 Sampling design and measurement**
To measure spatial heterogeneity and temporal changes in limnological parameters and methane fluxes, we
conducted three 36-hour surveys in summer 2019 (July 2-3, August 13-14, September 19-20). In the morning
(between 5-6 a.m.), we first measured surface values and vertical profiles of temperature, oxygen, and chlorophyll-
*a* concentration at the deepest point (see below for details). We subsequently installed 15 floating polyethylene
gas chambers (as shown in Fig. 1c), serving as fixed sampling sites and at the same time for accumulation of CH$_4$
fluxes (see further), starting in the western part of the fishpond. During installation (and further during each




sampling), temperature, pH, and oxygen concentration were measured at 0.3 m depth using the WTW 330i pH
meter and Oximeter (WTW, Weilheim, Germany). Vertical chlorophyll-*a* profiles were measured at each sampling
site using a submersible fluorescence probe (FluoroProbe, bbe-Moldaenke, Kiehl, Germany). From each site, the
average chlorophyll-*a* concentration in the surface layer (0-1 m depth) was used to assess the phytoplankton spatial
heterogeneity.
To minimise the chance that the differences observed among sites were due to time of day, we conducted repeated
measurements at the deepest point at the end of each sampling. If there was a change, all values were corrected for
the sampling time by linear interpolation:
$$P_{corr} = P_t + (P_{end} - P_0) \times \frac{(t - t_0)}{(t_{end} - t_0)}$$   (1)
where $P_{corr}$ is the corrected value of a parameter, $P_t$ is its value measured at the time t, $P_0$ and $P_{end}$ are parameter
values measured at the deepest point at the start (time $t_0$) and at the end ($t_{end}$) of the sampling. In the evening and
morning of the second day (roughly at 12 h intervals), we performed additional measurements of spatial
heterogeneity, allowing us to assess diurnal and nocturnal changes. In addition, samples for measuring $CH_4$
concentration in the surface water were collected at each site and analysed as described below. To assess diurnal
variations in thermal structure and oxygen concentration in the water column, we made vertical profile
measurements at the deepest point at 3-6 h intervals using the YSI EXO 2 multiparametric probe (YSI Inc., Yellow
Springs, USA).
**2.3 Methane measurements**
Water samples for determining $CH_4$ concentration in the surface water were collected at all 15 sampling sites in
triplicates into 20 ml glass bottles. The bottles were capped bubble-free under water with black butyl rubber
stoppers (Ochs, Germany) and sealed with aluminium crimps. Immediately after sampling, the water samples were
preserved by injecting 100 µl of concentrated sulfuric acid to stop the microbial activity (Bussmann et al., 2015).
The samples were processed within one week in the laboratory using a headspace technique according to
McAuliffe (1971). Methane concentration in the headspace was measured using an HP 5890 Series II gas
chromatograph (Agilent Technologies, USA) and calculated with the solubility coefficient given by Yamamoto et
al. (1976).
Methane diffusive fluxes (F) were then calculated for each sampling site indirectly using the 2-layer model with
the equation:
$$F = k(C_{sur} - C_{eq})$$   (2)





where $C_{sur}$ is the $CH_4$ concentration in surface water in µmol $L^{-1}$, $C_{eq}$ is the $CH_4$ concentration in surface water in
equilibrium with the atmosphere in µmol $L^{-1}$, and k is the $CH_4$ exchange constant (cm $h^{-1}$). The value of k was
calculated from the local wind speed according to Crusius and Wanninkhof (2003):
$$k = k_{600} \left(\frac{Sc}{600}\right)^n \qquad\qquad (3)$$
where $k_{600}$ is the gas transfer velocity for a Schmidt number (Sc) of 600. The Schmidt number for $CH_4$ was
calculated according to Wanninkhof (2014):
$$Sc = 1909.4 - 120.78t + 4.1555t^2 - 0.080578t^3 + 0.000658t^4 \qquad\qquad (4)$$
where t (°C) is the water temperature at the time of $CH_4$ extraction. The parameter $C_{eq}$ in Eq. (1) was determined
from the equation:
$$C_{eq} = \beta \times pCH_4 \qquad\qquad (5)$$
where β is the solubility coefficient of $CH_4$ as a function of temperature according to Wiesenburg and Guinasso
(1979), and $pCH_4$ is the partial pressure of $CH_4$ in the atmosphere.
To estimate total $CH_4$ fluxes from the water column to the atmosphere (i.e., diffusive and ebullitive fluxes), we
measured $CH_4$ accumulation in open-bottom floating polyethylene chambers (volume 3.1 L; area 0.024 $m^2$). Each
gas chamber was anchored at individual 15 fixed sampling sites, but allowed to float freely on the water surface.
Gas was accumulating for approximately 12 h during each sampling period, i.e., during the day and night periods.
Afterwards, 30 ml of gas was carefully taken from each chamber, after mixing the headspace in the chamber, and
stored in evacuated Exetainers® (Labco Limited, UK). Chambers were ventilated after each sampling period to
reset the incubation conditions. Methane fluxes were calculated as the difference between initial background and
final concentration in the chamber headspace and expressed on the 1 $m^2$ area of the surface level per day according
to Bastviken et al. (2004).
**2.4 Background limnological parameters**
During each campaign, samples for analysis of nutrient concentration and phytoplankton composition were
collected from the surface at the deepest point using a Friedinger sampler. Water transparency was measured using
a Secchi desk. Total phosphorus (TP) and soluble reactive phosphorus (SRP) were analysed
spectrophotometrically according to Kopáček and Hejzlar (1993) and Murphy and Riley (1962), respectively.
Concentrations of $NH_4^+$ and $NO_3^-$ were determined according to the procedure of Kopáček and Procházková
(1993) and Procházková (1959), respectively. Phytoplankton samples were preserved with Lugol's solution and





examined for species composition with an inverted microscope (Olympus IMT-2). Weather data were obtained
from the gauging station at the fishpond dam.

**2.5 Statistical analyses**

Two-tailed paired Student's t-tests and Two-way ANOVA with post-hoc Tukey's multiple comparison test (Prism
9.3, GraphPad Software Inc., La Jolla, USA) tested for differences between diffusive and total $CH_4$ fluxes between
day and night and among three sampling campaigns, respectively. The percentage of data variability explained by
different factors (daytime, month and site) was calculated with the Two-way RM ANOVA. Contour graphs
illustrating changes in spatial heterogeneity of measured parameters were constructed in Surfer 10 (Golden
Software, Inc., Colorado, USA) using the kriging contouring method. Spatial heterogeneity was quantified by
calculating the spatial variance (i.e., coefficient of variation):
$$CV\% = 100 \times \frac{SD}{mean} \qquad (6)$$
Higher spatial variance indicates increasing ecosystem patchiness. Linear mixed-effects models were used to
analyse the effects of $O_2$, pH, temperature, and water depth on the $CH_4$ diffusive fluxes with the random effect of
time of day nested within the effect of sampling date. The most parsimonious model was obtained by a manual
backward selection, where we sequentially removed all insignificant predictors ($p > 0.05$) using likelihood ratio
tests implemented in the drop1 function (Zuur et al., 2009). We also compared the slopes of the month-specific
regression lines produced by the model using analysis of covariance (Zar, 1984). Linear mixed-effects models
were implemented in the lme4 package version 1.1-21 (Bates et al., 2015), and Kenward-Roger F-tests were
computed using the ANOVA Type II function from the pbkrtest package version 0.4-7 (Halekoh and Hojsgaard,
2014). The prediction of the resulting final model was visualised in the package ggeffects version 0.14.1 (Lüdecke,
2018). Package performance version 0.4.4 (Lüdecke et al., 2020) was used to calculate Nakagawa's $R^2$ of the linear
model. The statistical analyses were performed using R software (v. 3.5.2, R Core Team, 2018).

**3 Results**

**3.1 Weather and background fishpond characteristics**

Weather parameters varied among sampling campaigns. In July, clear skies prevailed with the daily air temperature
above 30 °C (Table 1). During the August and September measurements, it was very cloudy, and daily air
temperatures decreased to 22 and 18 °C, respectively. The water level was stable during the whole studied period





with a monthly fluctuation of ~ 10 cm. Water transparency was low (15-40 cm), with an increasing trend towards
the end of summer (Table 1). Concentrations of total phosphorus and soluble reactive phosphorus were high (Table
1), consistent with a hyper-eutrophic state of the fishpond. In contrast, nitrogen concentrations were rather low,
with ammonium nitrogen being the predominant form of inorganic N in the water (Table 1).
**Table 1**: Basic characteristics of the Dehtář fishpond during the studied period.

|  | **July** | **August** | **September** |
|---|---|---|---|
| **Weather** | Clear sky, windy | Partly cloudy, no wind | Partly cloudy, no wind |
| **Air temperature (°C)** | 25-32 | 20-22 | 11-18 |
| **PHAR (mol m$^{-2}$ day$^{-1}$)** | 9.5 | 3.4 | 5.0 |
| **Secchi depth (cm)** | 15 | 30 | 40 |
| **TP (µg l$^{-1}$)** | 568 | 527 | 406 |
| **SRP (µg l$^{-1}$)** | 100 | 200 | 107 |
| **N-NH$_4^+$ (µg l$^{-1}$)** | 23 | 783 | 560 |
| **N-NO$_3^-$ (µg l$^{-1}$)** | 14 | 23 | 46 |
| **Chl-*a* (µg l$^{-1}$)** | 456 | 156 | 185 |
| **Phytoplankton composition** | Cyanobacteria | Cyanobacteria, green algae, cryptophytes | Cryptophytes, green algae |


Chlorophyll-*a* concentrations were highest in July due to the dense cyanobacterial bloom accumulated at the
surface (Table 1). The phytoplankton consisted of only three cyanobacterial taxa: *Dolichospermum flos-aquae,*
*Planktothrix agardhii,* and *Raphidiopsis mediteranea*. In August, phytoplankton was more diverse but also
dominated by cyanobacteria: *P. agardhii, Aphanizomenon issatschenkoi,* and *D. flos-aquae*. In September,
cyanobacteria were absent and instead, cryptophytes (*Cryptomonas reflexa*), green algae (*Pediastrum, Coelastrum*
and *Desmodesmus*) and dinoflagellates (*Ceratium hirundinella*) prevailed.
**3.2 Methane concetration and fluxes**
The CH$_4$ concentration in surface water was highly supersaturated over the whole studied period. The obtained
values varied from 0.003 up to 3.75 µmol L$^{-1}$ (Fig. 2), which corresponded to saturation levels of 108-12 834%. It
is obvious, that the obtained data show remarkable variance: the mean (± SD) values were 0.22 ± 0.18 for July,
0.34 ± 0.45 for August, and 1.61 ± 0.61 µmol L$^{-1}$ for September (Suppl. Fig. 11).





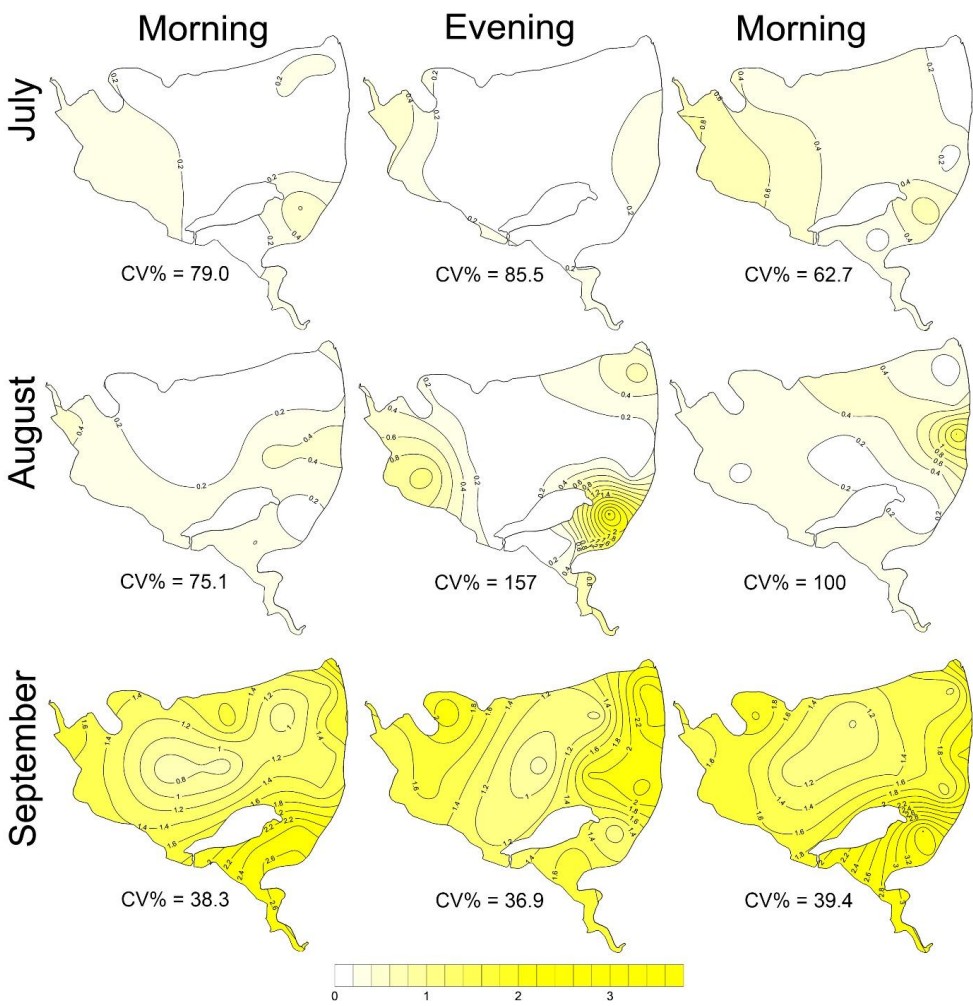

**Figure 2:** Contour graphs illustrating both seasonal and daily changes in spatial heterogeneity (indicated by the coefficient of
variation, CV%) of the surface methane concentration ($\mu$mol L$^{-1}$) in the fishpond.

Diffusive fluxes (i.e., calculated from CH$_4$ concentration, see Eq. 2) showed the lowest values in July and August
(average 0.12 and 0.16 mmol m$^{-2}$ d$^{-1}$, respectively) and pronouncedly peaked in September (average 0.78 mmol
m$^{-2}$ d$^{-1}$, Fig. 3a). By contrast, in July and August, the average total CH$_4$ fluxes (obtained with floating chambers)
showed the highest values (average 31.8 mmol m$^{-2}$ d$^{-1}$; ranging from 0.08 to 152 mmol m$^{-2}$ d$^{-1}$) while in
September, total CH$_4$ fluxes were three times lower than before (average 11.8 mmol m$^{-2}$ d$^{-1}$, range 0.3 to 43.5
mmol m$^{-2}$ d$^{-1}$, Fig 3b). As a result, diffusive fluxes accounted for only a minor fraction of total CH$_4$ fluxes to the
atmosphere (on average, 9.2 % in July, 4.1 % in August, 18.5 % in September, Fig. 3c).

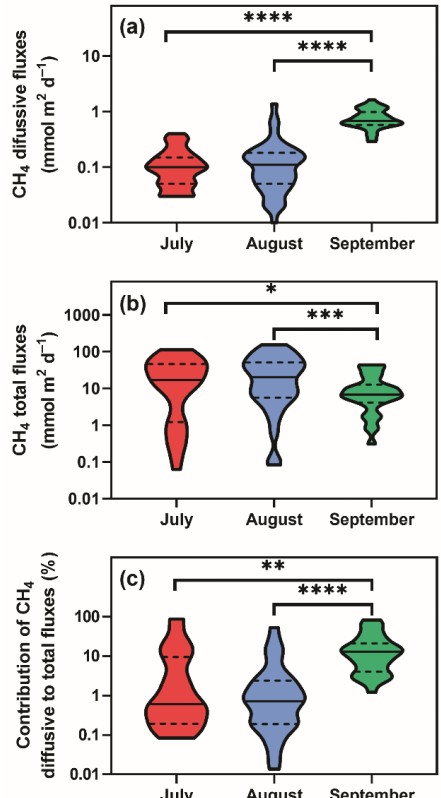

**Figure 3:** Violin plots of CH$_4$ diffusive (a) and total fluxes (b) during the studied period. Panel (c) depicts differences in the percentage contribution of diffusive to total fluxes. Solid lines are medians, while dashed lines denote quartiles. Asterisks indicate significant differences (* $p<0.05$, ** $p<0.01$, *** $p<0.001$, **** $p<0.0001$) between sampling dates determined by two-way ANOVA with Tukey's multiple comparison test. Note that a log scale is used here for clarity.

The total CH$_4$ fluxes show spatial variability within the fishpond that range several orders of magnitude (Fig. 3, 4; Suppl. Fig. 11; Suppl. Table 1). The observed spatial pattern showed high temporal variability on both daily and monthly scales (Fig. 2, 4, Suppl. Table 1). Most of the variability in CH$_4$ diffusive fluxes was explained by sampling date (62.4 %), while for the total CH$_4$ fluxes, spatial heterogeneity accounted for 87.2 % of data variability (Table 2). Using linear mixed-effects models, we identified water depth as the only significant predictor of total CH$_4$ fluxes (Df = 1, $p < 0.0001$, marginal Nakagawa's $R^2 = 0.348$; Fig. 5).



**Table 2**: The percentage of data variability explained by different factors (daytime, month = sampling date, and site)
calculated with the Two-way RM ANOVA. Statistical significant values ($p < 0.01$) are bold.

| | % of variability | | | | Significance | | |
|---|---|---|---|---|---|---|---|
| | **Daytime** | **Month** | **Site** | **Unexplained** | **Daytime** | **Month** | **Site** |
| **CH$_4$ diffusive flux** | 2.3 | **62.4** | 13.2 | 22.1 | 0.0123 | **<0.0001** | *n.s.* |
| **CH$_4$ total flux** | 0.19 | 2.4 | **87.2** | 10.2 | *n.s.* | *n.s.* | **<0.0001** |
| **pH** | **4.4** | **64.9** | 11.1 | 19.6 | **0.0001** | **<0.0001** | *n.s.* |
| **Water temperature** | **3.3** | **92.3** | **2.5** | 1.9 | **<0.0001** | **<0.0001** | **<0.0001** |
| **O$_2$** | **21.7** | **48.1** | **13.8** | 16.4 | **<0.0001** | **<0.0001** | 0.0135 |
| **Chl-*a*** | 0.019 | **74.9** | **16.7** | 8.4 | *n.s.* | **<0.0001** | **<0.0001** |

Interestingly, slopes of the linear regressions differed significantly among individual sampling campaigns (Fig. 5),
indicating an additional season-related factor that affects CH$_4$ fluxes in the fishpond. Calculated CH$_4$ diffusive
fluxes were not correlated with total fluxes. Linear mixed-effects models did not identify any significant predictor
of the fluxes, indicating that factors and processes out of the study's scope are involved. We found no significant
difference in either diffusive or total CH$_4$ fluxes between day and night.





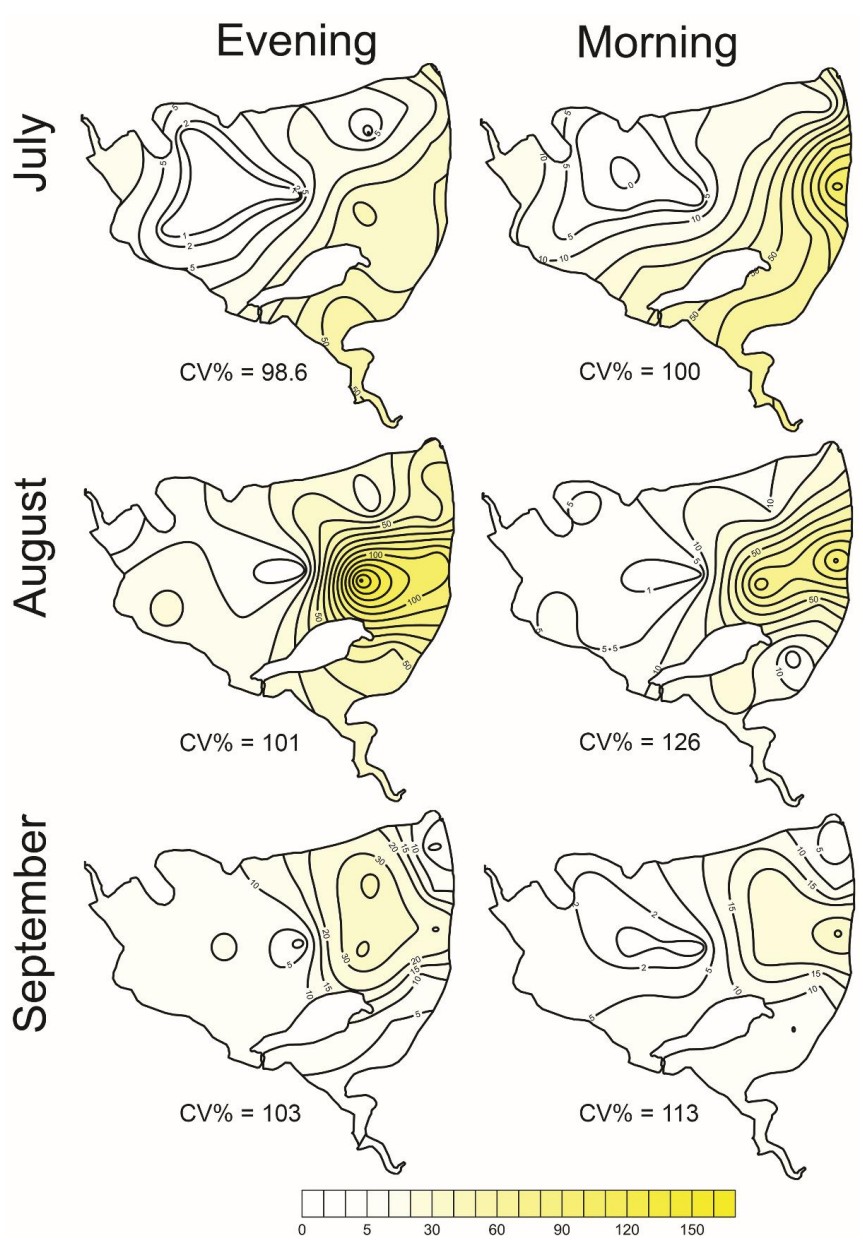


**Figure 4:** Contour graphs of methane total fluxes in the Dehtář fishpond. Isopleths connect sites with the same value of

methane fluxes (mmol m$^{-2}$ day$^{-1}$). CV% is a measure of spatial heterogeneity.



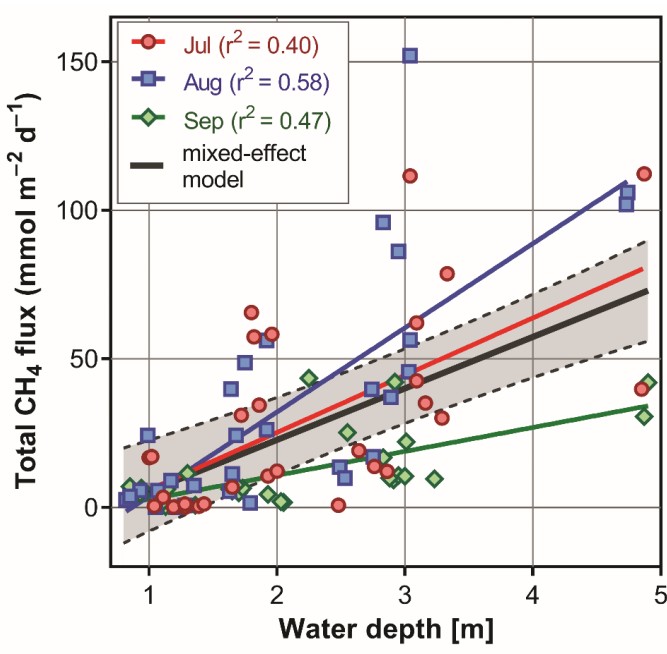

**Figure 5:** The most parsimonious linear mixed-effect model of methane total fluxes showing the water depth as the only
significant predictor. Symbols are the measured values, the solid black line is the prediction, and dashed lines are 95th
confidence intervals. Colours indicate month specific relation between total methane fluxes and water depth. Differences in
slopes were tested using the F-test. In September, the slope of the regression line was significantly different from that in July
and August.
**3.3 Diurnal changes in vertical profiles of oxygen and temperature**
Several contrasting patterns of vertical temperature and oxygen profiles occurred during summer 2019. Diurnal
changes were most pronounced in July (Fig. 6). Surface temperatures varied from 25 °C in the morning to nearly
30 °C in the afternoon. Thermal stratification of the water column was weak in the morning but became strongest
at 14:30 with a thermocline at 0.5 m depth (Fig. 6). Later in the afternoon, the water column began to be mixed by
wind. The morning vertical oxygen profile was characterised by a surface value of 4.3 mg L$^{-1}$, corresponding to
51 % saturation and anoxia below 3 m.

**Biogeosciences** Open Access
Discussions
EGU

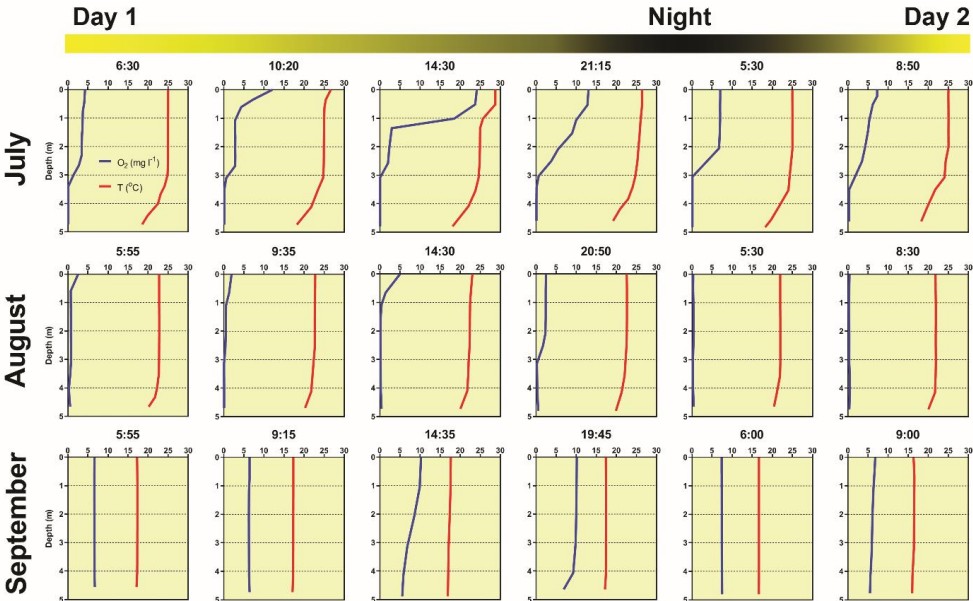

**Figure 6:** Diurnal changes in vertical profiles of temperature and oxygen concentration measured at the deepest point of the fishpond. Numbers above each graph indicate the time of measurement.

Due to the high photosynthetic activity of cyanobacteria, the surface oxygen concentration increased to 24 mg L$^{-1}$ (320 % saturation, Fig. 6), and a steep oxycline was established at a depth of 0.5-1.5 m with no effect on the anoxic conditions at the deeper layers. Wind action eroded both the oxy- and thermoclines in the evening, and by the next morning, the vertical profiles were similar to those at the beginning.

In August, the water column was almost entirely mixed and low in oxygen in the morning, with only 2.6 mg L$^{-1}$ (30 % saturation) of oxygen at the surface. Due to cloudy weather, the daily photosynthetic activity of phytoplankton resulted in only a slight increase in oxygen concentration at 0-1.5 m depth (4 mg L$^{-1}$, 47 % saturation). By the morning of the next day, the entire water column turned very close to anoxic (0.4 mg L$^{-1}$, 4 % saturation; Fig. 6), which in turn affected the spatial distribution of zooplankton, as evidenced by the formation of dense zooplankton clouds accumulated in the thin layer just at the surface (see Suppl. Fig. 3). In September, the water column was completely mixed, and we observed only weak daily changes in thermal and oxygen vertical structures (Fig. 6).

**3.4 Effect of wind on spatial heterogeneity of temperature, oxygen and chlorophyll-*a***

During the summer, all measured parameters showed remarkable within-lake spatial heterogeneity (Suppl. Fig. 4-8). In July, meteorological conditions allowed for demonstrating the effect of wind on fishpond spatial



heterogeneity. In the morning, there were no substantial differences in the surface temperature and oxygen
concentrations (Fig. 7ab). Phytoplankton biomass was accumulated mostly in the shallow western part, with the
maximum in the centre (Fig. 6c). At 14:00, a light breeze started to blow from the northwest, achieving a maximum
of 11 km h$^{-1}$ (Suppl. Fig. 7). This episode lasted till the evening measurement, and the wind ceased by 21:00. The
wind was strong enough to change spatial distribution substantially (Fig. 7d-f, Suppl. Fig. 4). In the evening, the
surface water temperature on the windward (south) side of the fishpond was ~ 4 °C higher than in the north (Fig.
7d). The wind also induced order of magnitude differences in oxygen concentration along the north-south axis of
the fishpond (3 mg L$^{-1}$ of $O_2$ at the north, 24 mg L$^{-1}$ of $O_2$ at the south; Fig. 7e) and affected phytoplankton
distribution in the fishpond, resulting in remarkable bloom accumulation in the south (Fig. 7f, Suppl. Fig. 8).
During the calm night after the disturbance, the north-south gradient substantially weakened. In August and
September, the thermal heterogeneity of the pond was rather low, but the spatial distribution of oxygen and
chlorophyll-*a* remained highly variable (Suppl. Fig. 5–8, Suppl. Table 1).

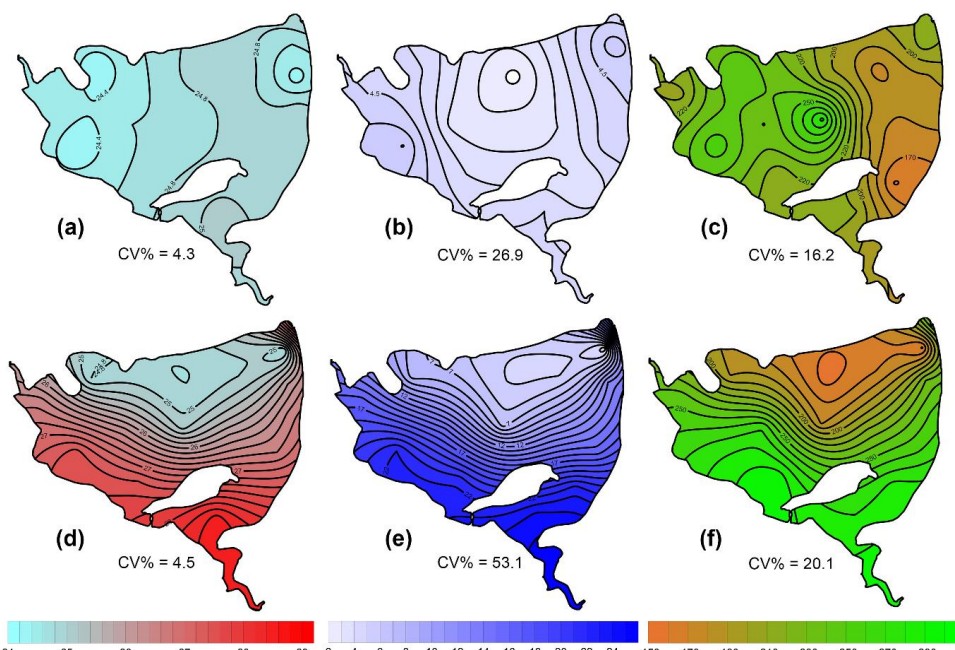

**Figure 7:** Contour graphs of surface temperature (a, d; °C), oxygen concentration (b, e; mg L$^{-1}$) and chlorophyll-a
concentration (c, f; µg L$^{-1}$) measured on July 2 at different times of day: a, b and c are the morning measurements; d, e and f
are evening measurements following a wind disturbance. Coefficient of variation (CV %) is a measure of spatial heterogeneity
of measured parameters.



## Discussion

### 4.1 Methane fluxes

Assessing spatial heterogeneity of the $CH_4$ fluxes within a fishpond is critical for a reliable estimate of its contribution to the global $CH_4$ budget. In our study, the variability in total $CH_4$ fluxes spanned several orders of magnitude (ranging from 0.06 up to 1 121.3 mmol m$^{-2}$ d$^{-1}$), which is in agreement with similar studies (Casper et al., 2000; DelSontro et al., 2016; Natchimutu et al., 2016). However, most system-specific $CH_4$ flux estimates rely on upscaling from a limited number of sites (Bastviken et al., 2004; Rasilo et al., 2015; Wik et al., 2016) because obtaining spatial variability in $CH_4$ emission is methodologically challenging. In general, spatial heterogeneity may reflect differences in water sources, physical mixing, local transformations and biogeochemical processes and rates among lake habitats (Loken et al., 2019). In deep lakes, littoral areas can contribute disproportionately to total lake $CH_4$ fluxes (Hofmann et al., 2010; Hofmann 2013, Natchimuthu et al., 2016; Schilder et al., 2013) and are often missed by traditional sampling approaches (Wik et al., 2016). According to Wik et al. (2016), low temporal and spatial resolutions are unlikely to cause overestimates. On the other hand, DelSontro et al. (2018b) suggested that horizontal transport of $CH_4$ produced in littoral zones and the interaction between physical and biological processes (e.g. air-water gas exchange, water column mixing, the interplay between $CH_4$ production and microbial oxidation) may result in an underestimation of whole-lake $CH_4$ fluxes based on centre samples. Similarly, Natchimuthu et al. (2016) found that up to 78 % underestimation would occur if samples obtained only from the lake center are used to extrapolate the total $CH_4$ flux. However, extrapolating our data from the deepest point of the Dehtář fishpond would lead to an overestimation of the $CH_4$ fluxes by a factor of 2.9 (Suppl. Fig. 12). The bias introduced by the deepest point measurement appears to be highly variable among systems with different morphology, geographical location, mixing regimes or trophic states. For instance, analysis of 22 European lakes during late summer has shown that spatially resolved $CH_4$ diffusive fluxes were highly variable for individual lakes, yielding 55–300 % differences in the whole-lake estimates (Schilder et al., 2013). Schmiedeskamp et al. (2021) observed an increase in $CH_4$ fluxes from the shore towards the centre in response to increasing sediment C-content in two shallow German lakes. In line with these findings, our results provide further evidence that spatially resolved data are needed to validate the uncertainties that come from using single-point samples to represent whole-lake processes in hyper-eutrophic systems. As stated by Loken et al. (2019), rather than assuming spatial homogeneity, scaling-up exercises of global carbon budgets should acknowledge the uncertainty that comes from extrapolating from spatially limited data sets.





In the Dehtář fishpond, the total CH$_4$ fluxes increased with water depth, and this relationship was month specific.
The highest CH$_4$ fluxes at the deepest points may seem contradictory to previous studies, in which the highest
fluxes were typically observed in littoral areas (e.g. DelSontro et al., 2018b; Hofman et al., 2010; Natchimuthu et
al., 2016; Schilder et al., 2013). However, these findings are based on studying mostly lakes whose morphology,
trophic state or oxygen regime sharply contrast with the Dehtář fishpond, where the upper two meters of the water
column were oxygen-saturated while the deepest strata were mostly anoxic. In such hyper-eutrophic systems, high
nutrient loading increases autochthonous primary production (Potužák et al., 2007; Rutegwa et al., 2019) and
promotes oxygen consumption and anaerobic decomposition in the sediments (Baxa et al., 2020), leading to
enhanced CH$_4$ production (Bastviken et al., 2004; Grasset et al., 2018). In aquaculture ponds in Southeast China,
CH$_4$ fluxes exhibited considerable spatial variations and peaked in the relatively deep feeding zone, where the
large loads of sediment organic matter fueled CH$_4$ production (Yang et al., 2020). Furthermore, sediment
temperature was the strongest predictor of CH$_4$ fluxes in ponds (DelSontro et al., 2016; Yang et al., 2020). It is,
therefore, reasonable to assume that both temperature and oxygen concentration in the sediment likely contributed
to changes in observed CH$_4$ fluxes during the studied period in our study. Although both parameters were not
directly measured in the sediment, it can be deduced from their vertical profiles that the probability of sediment
anoxia was highest in August and lowest in September, and the sediment temperature was lowest in September
(see Fig. 5).
Our results agree with the generally accepted view that processes other than diffusive fluxes – most likely
ebullition – represent the major CH$_4$ pathway to the atmosphere in hyper-eutrophic ponds (Kosten et al., 2020).
Although freshwaters with high primary production are more likely to have high CH$_4$ ebullition rates (DelSontro
et al., 2016), the dominant role of ebullition was also found across lentic systems differing in size, trophic status
or geographical location (Aben et al., 2017). Ebullition accounted on average for 56 % of total CH$_4$ fluxes in
northern ponds in Canada (DelSontro et al., 2016), 49 and 71 % in two different zones of Lake Taihu (Xiao et al.,
2017) and 48-83 % in three Swedish lakes (Natchimuthu et al., 2016; Jansen et al., 2019). The highest contribution
was found in the small hyper-eutrophic Priest Pot (UK), where ebullition represented 96 % of the total CH$_4$ flux
from the pond (Casper et al., 2000). Apparently, the contribution of ebullition can vary among systems and will
remain uncertain until measurement designs cover enough spatiotemporal variability to yield representative values
for the whole ecosystem.
In shallow water bodies, a semi-stable flux of microbubbles was suggested to account for a significant portion of
the total CH$_4$ flux (Prairie and del Giorgio, 2013). When CH$_4$ concentration in the water column is above a certain





threshold of microbubble density, these microbubbles likely aggregate, fuse, and escape to the atmosphere from
buoyancy (Prairie and del Giorgio, 2013). Even a small fluctuation in hydrostatic pressure (e.g., due to changes in
atmospheric pressure) or lake water level was shown to trigger enhanced $CH_4$ ebullition (Bastviken et al., 2004;
Casper et al., 2000; Varadharajan and Hemond, 2012). Since ebullition rates increase exponentially with
temperature, $CH_4$ fluxes tend to peak in warm summer months (van Bergen et al., 2019). In our study, 1 % lower
air pressure in July and August than in September, along with bottom anoxia and higher water temperature, could
account for the enhanced release of $CH_4$ bubbles from the sediment (31.7 mmol m$^{-2}$d$^{-1}$, >90 % of total $CH_4$ fluxes;
Suppl. Fig. 2). In September, when we observed the lowest water temperatures from the studied period and the
oxygen profile was rather uniform, ebullition accounted for 81 % (11 mmol m$^{-2}$d$^{-1}$) of the total $CH_4$ fluxes. The
spatially pooled data of the total $CH_4$ fluxes measured in the Dehtář fishpond varied from 11.8 to 34.5 mmol m$^{-2}$
d$^{-1}$, which is comparable with similar systems elsewhere (e.g., Bastviken et al., 2010; van Bergen et al., 2019;
Baron et al., 2022). To sum up, both diffusive fluxes and ebullition must be addressed to understand the magnitude
of total aquatic $CH_4$ fluxes and how their relative contributions vary across and within aquatic systems (Kosten et
al., 2020). Moreover, with an improved determination of $CH_4$ hot-spots and its causes, the management of ponds
could be changed accordingly and so the overall emissions reduced for example by decreasing P-availability and
dredging (Nijman et al., 2022).
**4.2 Effect of wind event on ecosystem spatial structure**
Sudden changes in ecosystem spatial structure in response to meteorological forcing have rarely been documented
(Loken et al., 2019) since they are hard to predict. Research into them using traditional methods requires intensive
effort or expensive instrumentation (Ortiz and Wilkinson, 2021), and it remains a matter of luck to obtain a relevant
dataset. In the July sampling campaign, we observed a strong impact of the wind on environmental heterogeneity
in the fishpond, which was apparent at a sub-daily time scale. Due to the methodological constraints, i.e., lack of
initial measurement, we can only speculate about the effect of wind on the total $CH_4$ fluxes. The northwest wind
during the day advected warmed surface water with cyanobacterial bloom from the north basin to the south. In the
evening, it resulted in bloom accumulation on the upward side and a north-south gradient of more than 4 °C and
4-24 mg L$^{-1}$ oxygen. After the winds fell off, the observed gradients declined during cooling at night. We assume
that the wind blowing across the pond surface drove buoyant cyanobacteria and surface water downwind and
caused an upwelling of deeper, colder, and hypoxic water on the upwind side. This wind-related circulation pattern
has been described as a "conveyer belt" in classical textbooks (Reynolds et al., 2006), held responsible for a
disruption of the thermal structure of the water column and the non-uniform spatial distribution of pH, oxygen,



$CO_2$ or $CH_4$ and also plankton assemblages (e.g. Loken et al., 2019; Natchimuthu et al., 2016; Rinke et al., 2009;
Ortiz and Wilkinson, 2021).
Similar to our study, mild winds (~4 m s$^{-1}$) were strong enough to redistribute heat and induce lake-wide
circulations driving upwelling and downwelling in 24 m deep Lake Pleasant (Czikowsky et al., 2018). As the wind
blows harder and lasts longer, the effects on ecosystem functioning may target higher trophic levels and become
more complex (Rinke et al., 2009). In Lake Constance, a three day storm event with wind velocities of ~10 m s$^{-1}$
resulted in a lake-wide displacement of water masses and the formation of the 6-15 °C horizontal surface water
gradient, which in turn changed the spatial distribution of phytoplankton, zooplankton and juvenile fish (Rinke et
al., 2009). After several stormy days (wind velocities of 12-15 m s$^{-1}$), Čech et al. (2011) observed negative effects
of wind-driven changes in water temperature and wave action on perch (*Perca fluviatilis*) spawning in the Lake
Milada. Although wind events affect shallow and deep lakes differently, there is growing evidence that they can
have far-reaching consequences on the functioning of aquatic ecosystems by disrupting energy flows, nutrient
fluxes, productivity and reproduction, and consequently altering community composition and trophic interactions
in the short and long term (Stockwell et al., 2020). As the frequency, intensity, spatial extent and duration of these
extreme meteorological events are projected to increase due to ongoing climate change (Comou and Rahmstorf,
2012), there is an urgent need to better understand the mechanisms underlying their impacts on the maintenance
of the ecosystem services.
**4.3 Summer changes in the oxygen regime**
Our data demonstrate that shallow, hyper-eutrophic ponds have disrupted oxygen regimes (Baxa et al., 2021) with
anoxic hypolimnion and may experience severe whole-water column hypoxia critical for aquatic biota (Miranda
et al., 2001). The hypoxic periods may result, for example, from sudden weather change (Jeppesen et al., 1990)
and last several days, during which physical processes and phytoplankton photosynthesis cannot compensate for
intense community respiration (Baxa et al., 2021). This became obvious in August when severe oxygen depletion
was measured at the surface across the whole pond, mostly far below a critical level of 4.5 mg L$^{-1}$, when adverse
effects came into play (Banerjee et al., 2019). However, oxygen surface concentrations in shallow parts of the
pond were substantially higher regardless of the time of day, which contrasts with findings of Miranda et al. (2001),
who emphasised shallow waters as the most sensitive parts of lakes, where hypoxic events can occur due to the
respiration of sediment biota. The observed spatial gradients of oxygen may create temporal refugia which allow
fish to survive harsh conditions that occur in the deepest part of the pond. To minimise economic losses and
negative impacts on the ecosystem, future research should identify the interplay between meteorological forcing,





trophic status and anthropogenic pressures (e.g. management practices) that affect oxygen fluctuations at various
time scales.
**4.4 Study limitations**
Like in other research, there are some limitations in the current study. Since our measurement had only a limited
temporal resolution (three samplings during the summer season), it is not appropriate to extrapolate $CH_4$ emissions
for annual values. Noticeably, future research must increase the frequency of the sampling and include also
innovative techniques to measure $CH_4$ fluxes at multiple fishponds, with different management regime. In our
study, the 12 h deployment time of the floating chambers could have led to extensive gas accumulation, which in
turn might have resulted in an underestimation of the total $CH_4$ fluxes due to the dissolution of the $CH_4$ from the
chamber into the water once the equilibrium concentration in the chamber is overcome (Bastviken et al., 2010).
However, $CH_4$ concentrations in water corresponded to a supersaturation of several orders of magnitude, so the
introduced bias appears to be of minor importance. In any case, our daily $CH_4$ fluxes represent a rather conservative
estimate for the global methane budget. In our study, we also did not address the important processes that could
shed light on the lake $CH_4$ budget, such as $CH_4$ oxidation rates (Bastviken et al., 2008) or biological interaction
(e.g. protistan grazing on $CH_4$ oxidising bacteria) in aquatic food webs (Sanseverino et al., 2012) that can affect
the overall $CH_4$ fluxes. We also lack information about spatial differences in sediment microbiota and organic
carbon content and compositions, which were found to affect $CH_4$ production rates (Berberich et al., 2020;
Emerson et al., 2021). Despite the limitation mentioned above, our results show that complementary spatial
surveys help contextualise the fixed station dynamics and provide additional, management-relevant information
about the fishpond.
**5. Conclusions**
Deciphering the mechanisms that drive spatial and temporal heterogeneity in aquatic ecosystem structure and
function not only expands our understanding of pond ecology but also provides insights to improve the
management of these ecosystems and the services they provide. Our results suggest that spatial heterogeneity needs
to be considered when designing experiments and monitoring programs. Without the spatially resolved sampling,
we introduce bias into our datasets, hampering our limnological understanding of the ecosystem's functioning and
impeding our ability to accurately estimate rates such as methane emissions on a global scale (DelSontro et al.,
2018a). In agreement with Kosten et al. (2020), we demonstrated that neglecting ebullition leads to a considerable
underestimating of the total $CH_4$ fluxes. Since there are thousands of these intensively managed fishponds, we



argue for changing the management practices toward sustainable use of natural resources to mitigate the overall
emissions of greenhouse gases from these ecosystems. Future studies are needed to characterise $CH_4$ fluxes over
a greater number and diversity of aquaculture ponds and examine the mechanisms controlling $CH_4$ emissions in
aquatic ecosystems.
**Acknowledgements**
The study was supported by the Czech Science Foundation (Research Projects No. 17-09310S, 19-23261S and
P504/19-16554S). We thank Dr. Martin Rulík for providing us gas chambers. We especially thank to Prof.
Miloslav Šimek and Linda Jíšová for enabling gas analyses. We are grateful Anna Sieczko for consultation on the
calculation of $CH_4$ fluxes. English correction was made by Anton Baer.
**Data availability**
Dataset associated with the manuscript can be found in the GitHub Repositories under
https://zenodo.org/badge/latestdoi/587640213.
**Author contributions**
All authors contributed to the study conception and design. PZ planned the campaign; PZ, AM and JN performed
the sampling and analyzed the data; AM performed the gas-measurements; VK performed statistical analyses and
modelling; PZ and AM wrote the manuscript. All authors read and approved the final manuscript.

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
