# Peer review of "Spatial and temporal variability of methane emissions and"

_Biogeosciences, 2023_

## Referee Comment (RC2)

[referee-annotated manuscript omitted]

---

## Author Response (AR1)

**Referee1**

**At the end of the manuscript, I expected a paragraph regarding recommendations for improved monitoring strategies for better estimation of methane emissions. The paper would gain a lot if all the recommendations made in the text were consolidated in one place in terms of best practice actions: how to measure, where, who often, which additional information is needed, ….**

As suggested, we added a separate paragraph with our recommendations at the end. For clarity, we inserted it right here.

"For improved monitoring strategies, however, a continuous measurement approach like eddy covariance is generally more efficient than traditional sampling at regular intervals. Eddy covariance accounts for temporal variability and provides high temporal resolution data by continuously measuring wind speed, gas concentration, and vertical turbulent fluxes to estimate methane emissions (Erkillä et al., 2018). More importantly, it also offers spatially integrated measurements, averaging emissions over a larger area and therefore accounts for pond spatial heterogeneity. However, it's worth noting that the choice of monitoring approach depends on various factors, including the specific objectives, available resources, and the characteristics of the emission sources. To accurately capture both short-term variability and lake spatial heterogeneity of methane ebullition and diffusion fluxes, the most efficient approach was found to be a combination of continuous measurements with traditional methods including floating chambers, anchored funnels and boundary model calculations (Schubert et al., 2012, Podgrajsek et al., 2014, Erkillä et al., 2018). This integrated approach would provide a comprehensive understanding of methane emissions, enabling better estimation and more effective mitigation efforts."

**Some detailed comments:**

**Line 48-49 – Sanseverio et al., 2013 – in reference list 2012?**

Corrected to 2012.

**Line 70 – Jansen at al., 2020 – in reference list 2019?**

Corrected to 2019.

**Line 94 – 96 – you measure the values at the deepest point (at each sampling point or the deepest point of the pond?). It is not clear to me at this point of the manuscript, later in text it becomes more clear. However, it would be good to mark this point in Fig. 1c as a prominent sampling point.**

Done, see a new version of Fig. 1c and the description.

**Line 101 – bbe_Moldaenke, Kiel**

Corrected.

**Line 202, Fig 2: As a comment – maybe the CV is not so well suited to show spatial heterogeneity of CH4 concentration. The mean values for July and August are very low with high SD, resulting in high CV values?**

We used the CV as a commonly used simple and easily interpretable parameter describing variability. As the value of CV is normalized for mean (CV=100 x standard deviation / mean) it allows comparing of different variables which may attain different numerical (mean) values. It reflects RELATIVE variability of values observed at different sampling points. Values in July and August are generally low but unequal, so CV is high. In September, values are high but more evenly distributed so CV is lower. Therefore, in our opinion, the used of CV is OK.

**Line 479-481 – Beaulieu et al. not mentioned in main text**

Deleted from the reference list.

**Line 577-579 – Ostrovsky et al. not mentioned in main text**

Deleted from the reference list.

**Line 623-628 – Tranvik et al. not mentioned in main text**

Deleted from the reference list.

**Comments:**

**Introduction: clarification which parameters react on which time scale…. And why these parameters are important for the spatio temporal variability…?**

We tried to comply with the comment:

"The complex interactions between physical and biological factors lead to a dynamic and ever-changing environment, characterised by high spatial and temporal variability of methane fluxes in ponds."

**L 107 ff: it would be interesting to see the variations at one fixed station during the day!**

We agree. This could be partially deduced form Fig. 5 showing diurnal changes in the whole water column at the deepest point of the pond. We added additional Suppl. Fig. 13 (fig. answ1) to the Supplementary material demonstrating diurnal changes of oxygen in the first campaign in July, where the diurnal variation was highest.

[Figure]

**How long did it take to set up all the sampling sites?**

This is a good point. We added a specification how long did it take to set up all sampling sites in the Material and Methods. Briefly, the initial set up of all sampling sites (i.e., deployment of all floating chambers) took almost 4 hours (3:50). Unfortunately, during this initial phase of our measurement, dial change in e.g., oxygen was substantial (see a new Suppl. Fig. 13) and this was actually the reason to correct the measured parameters to compensate for the increase. All other measurement took only 2 hours. Moreover, in other dates, the daily changes the parameters were negligible, see Fig. 5.

**The correction of the parameters for temporal variations is based on the assumption that the fluctuations are linear between the measured time points? But I would assume that at least oxygen (or parameters related to photosynthesis) vary with a sinus curve….**

This is related to our explanation above. The daily curve of e.g., oxygen variation may have the unimodal shape, as demonstrated in a new Suppl. Fig. 13. However, our correction was done only for the duration of all-site sampling, (2 hour only), i.e., to compensate how a parameter changed between the measurement at the first and the last sampling point each morning, evening and next-day morning. For such a short period, linear approximation is the most suitable. Moreover, with exception of the initial measurement, during which we had to install all the floating chambers, the correction was negligible. We added this information to Material and Method.

**If all data were corrected for this temporal variation, how can you assess this influence later on with the statistics, as daytime variation?**

This was explained earlier in our comment.

**The sampling scheme to assess the temporal and spatial variations are not well explained, and confusing…? May be a table would help here, with the deep station compared with the other stations.**

In the revised version of the manuscript, we specified some important but originally missing details. Moreover, adding the markers to all contour graphs in the revised manuscript should improve the clarity of the sampling scheme.

**L125 ff, The Calculation of the diffusive flux is missing some crucial parameters: how was k determined? Which atmospheric CH4 concentration was used?**

We have clarified the description in the revised version.

To compute $k_{CH4}$ values we first derived $k_{600}$ estimates using a wind speed-based relationship according to Crusius and Wanninkhof (2003):

$$k_{600} = 1,68+(0,228*U_{10}^{2,2})$$

where $U_{10}$ represents the wind speed at 10 m height (in $m.s^{-1}$; obtained from the nearby meteorological station) approximated by $U_{10} = 1.22U$, where U is the measured wind speed at 1.5 m height. We then converted $k_{600}$ to $k_{CH4}$ using the eq. 3.

The atmospheric partial pressure of $CH_4$ was set to 1.8 ppm; we have specified that in revised version.

**Discussion:**

**L310 ff: Could you give an estimate how many stations would be needed for your lake to get a good coverage of the variability? Could you calculate and plot the CV for n = 3, n=4…. n=16 samples**

To assess the number of stations needed to get good coverage of variability of different variables, we randomly selected different numbers (n=3-15) of stations and calculated CV for different variables, we repeated this 100x for each variable and plotted the result (fig. answ2 exemplifying this for $O_2$ data), then we calculated the error (CV%) and found the station where this error dropped under 10% (fig answ3). We repeated this 2x for each variable and get the following site numbers: 11 and 14 for $CH_4$ concentration, 12 and 13 for $CH_4$ flux, 12 and 13 for $O_2$ concentration, 11 and 11 for temperature, 13 and 14 for temperature. Although we have chosen the number of 15 stations intuitively, this number seems to be sufficient – providing 10% is an acceptable error.

[Figure]

Fig. answ2. For detail description, see the text above.

[Figure]

Fig. answ3. For detail description, see the text above.

**The results are well related to other studies, however explanations or reasons for the obtained results in this study are missing.**

With all due respect, we disagree with this comment. We can only speculate about the observed spatial patterns of the environmental parameters. The high patchiness, i.e., deviation from uniformity is generally assumed in hyper-eutrophic shallow ponds, but the fact is rarely addressed in the scientific literature. In our study, we were lucky enough to describe the effect of wind action on spatial patterns during the first sampling in July. The mechanisms underlying, for instance, the effect of the depth on spatial heterogeneity in methane fluxes are discussed, please see lines 325-339. Nonetheless, it is fair to say that we could not always find plausible explanations for all our findings, which we admit in study limitations.

**L316 ff: Why was the highest CH4 flux in your lake at the deepest station? Could you give any ideas?**

Several factor beyond the study scope might be involved, which are mentioned in Study limitations, e.g., sediment organic carbon quality and quantity, microbial community dynamics. In Discussion, we speculate about the role of the extent and duration of bottom anoxia (explicitly specified in the revised version on lines 327-329).

**L352: why did you find highest ebullition rates in September in your lake?**

In fact, we did not find the highest ebullition rates in September, but the highest $CH_4$ concentration in the water column, which lowers the contribution of ebullition to the total $CH_4$ fluxes in September (Fig. 3). We speculate about a possible role of temperature on seasonal differences in methane concentrations, see lines 343-346.

**Notes from the pdf file**

**L46 why are the following parameters important for the spatial-temporal variability? Please elaborate...**

We expanded the text:

"The complex interactions between physical and biological factors lead to a dynamic and ever-changing environment, characterised by high spatial and temporal variability of methane fluxes in ponds."

**L47 but eutrophication and water depth do not react on short time scales......**

We replaced eutrophication with more appropriate nutrient loading, which may change dramatically in response to sudden weather events, e.g., heavy rainfalls. Water depth was removed.

**L71 their**

Has been corrected.

**L105 it would be interesting to see the variations at one fixed station during the day..! If all data were corrected for this temporal variation, how can you assess this influence later on with the statistics, as daytime variation?**

We already responded, see our earlier comments.

**L130 was the wind measured or from a database?**

The data on wind speed and direction were obtained from a gauging station situated right at the pond dam.

**L131 how was k determined / calculated? Equation 4 only describes the conversion from k-600 to k-CH4. which value for n?**

See previous answer to this question (L125).

n is a wind speed-dependent conversion factor, for which we used $-2/3$ for $U_{10} < 3.7$ m s$^{-1}$ (Jähne et al., 1987); we have specified that in revised version.

**L140 did you also determine k from the chamber measurements? to compare it with values calculated?**

We did not estimate k from the floating chamber measurement. The floating chambers in this study were without an ebullition shield so they received both diffusive and ebullitive fluxes, which in turn leads to much larger k values than those estimated simply by model-based calculations. Hence, the total flux was calculated by dividing the change in the $CH_4$ partial pressure in the chamber headspace with the duration of chamber´s deployment (incubation) time and the chamber area as described in Bastviken et al. (2004).

**L143 this is a rather long incubation time...?**

It may seem so, but it depends on the context. Usually, incubation times from minutes to hours (one day) have been used in similar studies depending on the study site and experimental design. The scheduling of methodology was based on studies that compared short-term (several minutes) vs. long-term (24 h) deployment of the gas chambers (Bastviken et al., 2004; 2010). We are aware of possible bias from gas accumulation in the chambers during a longer period, what we also discuss in the manuscript (see L411-417). However, the bias can be compensated for in the calculations as we know the ambient $CH_4$ concentration and water temperature.

**L146 thus for each incubation there was a start and end point?**

That is correct, we specified that in revised version.

**L150 water samples?**

Has been corrected.

**L152 disk**

Has been corrected.

**L187 wind speed should be added here; would water temperature not be more appropriate?**

Done.

**L198 all stations, all sampling dates?**

Basic characteristics of the Dehtář fishpond during the studied period, measured at the surface at the deepest point, see L162-163 and a new Table 1 description:

"Basic characteristics of the Dehtář fishpond during the studied period, measured at the surface at the deepest point."

**L202 please add markers for the stations**

Done in all figures including those in the supplementary material.

**L203 I do not understand how the cv was calculated, based on mean ± std per day? per sampling ?? "surface methane concentration" should be mentioned at the beginning of the legend.**

We calculated CV per sampling using the 15 values measured at 15 sampling sites. Text amended to clarify this, see also our earlier comments.

**L213 panel c does not seem necessary to me**

It may seem to be redundant, but we consider having all relevant information on the same panel. Since it does not require much space, we prefer to leave the figure as it is.

**L228 check which number have to be in bold, for pH, temp, O2**

Done.

**L272 convert to SI unit, m/sec**

Done.

**L289 how many ("several orders")?**

Done.

**L304 what might be the reason for this overestimation?**

The overestimation comes from the methane flux depth-relation. If the methane flux is the highest at the deepest sampling point, thus the spatially-pooled values accounting for spatial heterogeneity, i.e., more reliable methane flux estimation, are lower.

**L316 what is the reason for this for your lake?**

We have already answered the comment, please, se our responses above.

---

## Author Response (AR2)

We would like to thank all reviewers for their time and effort in improving the overall clarity and quality of the original draft manuscript. Since Rev#1 does not propose any additional changes, we address the comments and suggestions made by Rev#2.

The main point raised by the reviewer is to improve the overall framework of the manuscript to include, for example, details about management practices in the fish ponds studied. Therefore, we expanded the study description to include fishpond management in more detail (lines 78–82). We also added a short paragraph to the conclusions section putting our results in a broader context (see lines 451–460). As suggested, we went through the text carefully and made technical changes to standardise units, unify formatting or correct typos.

Specific comments (our reply is red):

48-49, After reading the references cited here, I remain skeptical that phytoplankton are themselves producing methane, please qualifying this statement by reframing with an appropriate term such as "preliminary" or "potential".

Done.

76, Missing digits in Lat/Long?

Added.

324-330, It is unclear which ponds are being referred to in these two sentences. Upper layers being O2 saturated and sediments being anoxic does not consistently fit the data from Fig. 6, but the next sentence starts with "In such [systems]".

That's a good point. However, the confusion may rise only in September, where oxygen concentration in the water layer right above the bottom was around 6-7 mg L$^{-1}$, corresponding to hypoxic conditions but not to anoxia. Noteworthy, when the oxygen probe was accidentally buried in the sediment, oxygen concentration went rapidly low down to zero. Therefore, we believe that even in September, there was anoxia in the sediment with a steep gradient at the water-sediment interface. To avoid confusion, we start the sentence with „In hyper-eutrophic systems"

333, Here too, which kinds of ponds?

Specified, see line 339.

339 – 349, Is it that ebullition is the dominate pathway in hyper-eutrophic ponds, or most small water bodies?

In the paper referred to, the statement is related to the trophic state but not directly to the size of the system. Since the size may vary significantly among the systems, we, therefore, specify the aquaculture production rather than the size of the ponds, see line 346.

377, Is 4 to 24 mg L-1 O2 meant to be the gradient here? If so, "…a gradient of more than 4°C and 20 mg L-1 O2" would be clearer.

Done.

424, Be specific about your estimate here, what is the value?

Done, see lines 429–430.

432-444, formatting issue

Checked and corrected